# Psychometric Properties of the Multidimensional Body–Self Relations Questionnaire—Appearance Scales (MBSRQ-AS) in Chilean Youth

**DOI:** 10.3390/ijerph20010628

**Published:** 2022-12-29

**Authors:** Paula Lizana-Calderón, Jesús M. Alvarado, Claudia Cruzat-Mandich, Fernanda Díaz-Castrillón, Sergio Quevedo

**Affiliations:** 1Centro de Estudios de la Conducta Alimentaria (CECA), Escuela de Psicología, Universidad Adolfo Ibáñez, Av. Diagonal Las Torres 2640, Santiago 7911328, Chile; 2Facultad de Psicología, Universidad Complutense de Madrid, 28223 Madrid, Spain

**Keywords:** multidimensional body–self relations questionnaire—appearance scales (MBSRQ-AS), exploratory structural equation modeling (ESEM), body image (BI), mimic models, body mass index (BMI)

## Abstract

The aim of this study is to analyze the factorial structure of the Multidimensional Body–Self Relations Questionnaire—Appearance Scales (MBSRQ-AS) to determine the adjustment of the study structure proposed in the Spanish short version in a young, non-clinical Chilean population and to evaluate the possible influence of sex, age, and BMI on body image measurement. The sample consisted of 614 Chilean youth (259 male and 355 female) between 15 and 28 years old (M = 18.81; SE = 2.46), from the Metropolitan Region, and four regions of the coast and south-central zone of the country. The average Body Mass Index (BMI) was 22.5 kg/m^2^ (SD = 3.16). The model fit was evaluated by confirmatory factor analysis (WLSMV) using the following: a model with a single general factor, a model with the five factors of the original version, a five-factor ESEM model, and a MIMIC model analyzed including sex, age, and BMI. The results show that the MIMIC sex, age, and BMI model presents an acceptable fit, observing that four factors, Appearance Orientation (AO), Body Areas Satisfaction (BAS), Overweight Preoccupation (OP), and Self-Classification Weight (SCW), are affected significantly for the sex variable, one for age, Appearance Evaluation (AE), and four for BMI (AE, BAS, OP, and SCW). In conclusion, MBSRQ-AS replicates the five-dimensional structure in a non-clinical sample of young Chileans; however, their scores are not invariant as they depend on sex, age, and BMI.

## 1. Introduction

The concept of body image (BI) has acquired importance due to the relationship between body dissatisfaction and eating disorders (ED) as well as physical self-concept [1]. Additionally, it is a theoretical construct used in various fields such as psychology, psychiatry, and sociology to explain aspects of personality such as self-concept or self-esteem and various psychopathologies [2]. The BI corresponds to the mental representation that each individual builds in terms of feelings, behaviors, and attitudes in relation to their own body; therefore, a perceptual image, a cognitive image and an emotional image about the body stand out [2,3,4]. This is a multidimensional construct that includes different factors, encompassing aspects such as the perceptual component, referring to the perception of the size of body segments related to the concept of mental body schema; the subjective component, which is related to cognitive and affective aspects of the assessment of the body or its parts, including thoughts, beliefs, and our feelings about the degree of satisfaction with one’s own body; and finally a behavioral component, referring to specific behaviors that individuals perform regarding the consideration of the shape of the body and the degree of satisfaction with it and that could be focused on to improve their appearance, their health, their physical condition, or their weight [3,5,6,7].

Numerous studies have been carried out on the relationship between body image and eating behaviors, especially in adolescents and in non-clinical samples, for preventive purposes in relation to the appearance of certain problems such as eating disorders. Spanish studies [8,9] concluded that adolescent women have lower body satisfaction than young men, identifying significant differences in body dissatisfaction, this being higher in women during early adolescence. An Argentine study highlights the predictive value of dissatisfaction with body image in relation to the presence of eating disorders in adolescents [10]. A Chilean study [11] with 437 Chilean adolescent women identified a significant relationship between pressure from social agents to achieve an ideal of slim beauty and the presence of altered eating attitudes and behaviors; this pressure from social agents emerges as a strong predictor of symptoms of eating disorders. Added to this, another Chilean study on the body image of young Chileans with normal weight and overweight/obesity identifies young women with a more negative evaluation of their body image compared to men [12].

On the other hand, studies have also been carried out on the relationship between body image and real and perceived body mass index (BMI) and obesity, where there was a direct and significant correlation between BMI and body dissatisfaction [8]. A longitudinal study found that overweight young women with a lower level of body satisfaction had a greater increase in their BMI over 10 years compared to young women with high body satisfaction [13]. These results are consistent with a study of 376 young Chilean students [12] where the overweight/obese group presents greater dissatisfaction with their body image, a worse diagnosis of health and lower self-assessment of their physical condition and greater concern and self-classification for their own weight. In comparative studies, they found that Chilean students present a greater overestimation of their weight than Panamanian and Guatemalan students [14].

Other Latin American studies show the negative association between weight and body dissatisfaction. In Brazil, female adolescent students presented higher body image dissatisfaction than males, and body image dissatisfaction was higher among adolescents with overweight or obesity in both sexes [15,16]. This coincides with what was investigated in Chile, in which males and females from rural areas who are overweight or obese underestimate their body weight [17]. Additionally, in research with university students from different localities or cities of the country, a low concordance between the real and perceived nutritional status is observed; obese students tend to underestimate their weight, underweight students overestimate it, and underweight and normal-weight women overestimate their actual weight even more [18]. In Spain, a study conducted with 935 young people aged 10–18 identified that dissatisfaction with body image increased with age, especially in females; the means of dissatisfaction with body image were higher in women than men and differences between sexes increased with age [19].

The evaluation of this construct becomes important in adolescence, a period during which the body becomes a fundamental aspect of identity development. Therefore, the evaluation and orientation of appearance, that is, the feelings of satisfaction with the appearance that we assign to how we see ourselves and the attention to one’s own appearance, have an impact on body image and self-concept. In the adolescent stage, both young men and women attach great importance to appearance, realizing that physical appearance is a significant aspect of adolescent identity [20]. In this way, the development of body image constitutes one of the most important psychological tasks of adolescence [21] to the point that the simple fact of being an adolescent woman, as a group, is associated with very low levels of satisfaction while overweight women have even lower levels, which is related to poorer levels of self-care in terms of healthy behaviors [22].

This is linked to the development and construction of beauty ideals, under the idea that body image is socially determined by social influences [2,23], which in turn influence the different levels of personal satisfaction/dissatisfaction with one’s own body [24,25]. The concept of body image is not fixed or static but is a dynamic construct that varies based on personal experiences and social influences [2,23], and its relationship with sociocultural standards present in certain times and cultures would affect the degree of satisfaction towards the body [26,27], making adolescent women especially vulnerable to the ideal of thinness [15,28,29]. This ideal of thinness would constitute a powerful risk factor for the generation of an unsatisfactory body image [30,31] and the development of eating disorders [32,33].

In this line, research has been carried out on the measure of satisfaction of people with their body image, in association with the development of identity, self-concept and self-esteem. Preliminary evidence suggests that the process of identity development in adolescence is related to body image [34] This is how Voelker et al. [34] concluded that the evidence would support that body image may improve in adolescence as the sense of self stabilizes. On the one hand, when higher levels of stress associated with identity issues are observed, a strong relationship is established with variables such as appraisal of appearance, satisfaction with body areas and concern about weight [20]; therefore, the degree of concern with identity issues would be a predictor of evaluation by appearance.

Studies maintained that individuals with high body dissatisfaction and with negative representations of themselves had lower levels of self-esteem, less clarity in their self-concept and greater negative affect, and high levels of self-esteem are associated with lower levels of body dissatisfaction [29,35]. In addition, Moradi et al. [36] carried out a systematic review regarding children and adolescents with overweight or obesity and their relationship with depression, anxiety, low self-esteem, and body dissatisfaction, highlighting that the findings show a positive relationship between the risk of body dissatisfaction with obesity in adolescents and children, as well as a direct and significant relationship between obesity and the risk of low self-esteem.

It is also relevant to acknowledge the negative implications of body image on mental health. For instance, psychiatric disorders had the greatest association with negative body image and body dissatisfaction can predict body-image-related psychopathology later in development [37,38]. For instance, depressive symptoms are related to the wish to be thinner; in other words, body dissatisfaction is associated with increased depressive symptoms [39]. On the other hand, positive body image could have significant implications for well-being and mental health; thus, having higher levels of positive body image may be a protective factor for depression and poor self-esteem [40].

Based on the above and in relation to the implications of body image, its consequences, and its effect on mental health, it is necessary to have integrative instruments that allow us to understand body image precisely from a multidimensional approach. One of the most used instruments for the evaluation of body image is the Multidimensional Body–Self Relationships Questionnaire (MBSRQ) [41], which has been used since its inception in many studies associated with the subject of weight, and in other areas such as alopecia, acne and the effectiveness of therapies focused on body image [42] as well as in university students with normal weight, obesity, eating disorders and physical exercise. It is considered one of the most used instruments to measure body image [43,44], specifically the evaluation of body image attitudes [45]. Comprising a self-report of 69 items expressed in 10 subscales, in its evaluation, the questionnaire includes evaluative, cognitive, and behavioral components and evaluates attitudinal aspects of body image. Regarding its dimensions, the questionnaire includes the evaluation and orientation towards health/disease and physical exercise, as well as the evaluation of their weight and the dimensions of satisfaction and dissatisfaction with appearance, thus presenting seven main factors and three subscales. It is designed for application in people aged 15 years or older, being a multidimensional instrument used in research on body image, from basic psychometric studies to clinical and applied research [41], with evidence of validity in community samples of men and women [45].

From the interest of researchers in body image, specifically in appearance, a limited version of the inventory is generated, the Multidimensional Body–Self Relationships Questionnaire—Appearance Scales (MBSRQ-AS) [41], composed of 34 items and 5 subscales that measure the orientation and evaluation of body image with a focus on body appearance. It excludes the items of orientation and evaluation of physical condition and health that the complete version of the questionnaire has. The MBSRQ-AS consists of two factors from the original inventory and its three additional scales. The subscales of the MBSRQ-AS are Appearance Evaluation (AE), Appearance Orientation (AO), Body Areas Satisfaction (BAS), Overweight Preoccupation (OP), and Self-Classified Weight (SCW). For instance, this version is recommended when the objective is to evaluate appearance related to body image, and there is no interest in health scales.

Numerous adaptations and validations of the questionnaire have been made in its 34-item MBSRQ-AS version, into German [46], French [47], Greek [48], Pakistani [49], Malay [50], Spanish [51] and in the Latin American context, specifically in the Brazilian [52] and Mexican population, both in the general population [44] and only in men [53]. The Spanish version of the MBSRQ-AS [54] was tested in a non-clinical sample composed of 1041 people between 15 and 46 years old, confirming that the Spanish version of the MBSRQ-AS has the same five factors reported by Cash [41]. Furthermore, using the same Spanish version, a study comprising 355 adolescents between 12 and 14 years of age concluded that the factorial structure of the five factors of the original inventory was confirmed [51]. The Pakistani version was tested on a sample of 850 adults, and, from an exploratory and confirmatory factor analysis, four factors were concluded: body area satisfaction, appearance orientation, appearance evaluation, and concern about being overweight [49]. The German study tested the psychometric properties and the factorial structure of the German translation of the MBSRQ-AS [46] in 230 female patients with eating disorders and 293 healthy women, highlighting that the subscales of the questionnaire showed good reliability and convergent and discriminant validity coefficients. The study of the psychometric properties of the French adaptation of the MBSRQ-AS was carried out in 765 people between 18 and 61 years old, considering only two subscales: appearance orientation and appearance evaluation, concluding the same two factors as the original measure [47]. Similarly, the Greek study was conducted with 1312 school students and factor analysis revealed that the Greek items of the MBSRQ-AS were significantly loaded with the main factors of the scale [48]. The psychometric properties of the Brazilian adaptation were tested with a sample of 1005 participants, highlighting that this version has the same factorial structure as the original MBSRQ-AS [52]. Likewise, adaptations and psychometric analyzes of the Persian version have been carried out in 251 Iranian women with polycystic ovary syndrome [55,56], with the aim of translating and evaluating the psychometric properties of the inventory, concluding from a confirmatory factor analysis a good fit index for the five factors, thus confirming the factor structure of the MBSRQ-AS. Additionally, exploratory factor analysis studies of the MBSRQ-AS inventory with a Malaysian population of 629 adults found a reduction in the four dimensions, even though one factor had less-than-adequate internal consistency; therefore, this factor was omitted, resulting in a solution of 3 factors and 23 items [50]. From a confirmatory factor analysis, the authors maintained that both models (five-factor model and three-factor model) had a good fit in some indices (RMSEA < 0.08) but a less-than-ideal fit in three indices (robust CFI and robust TLI < 0.9; SRMR > 0.06), showing that the three-factor model obtained a comparatively better fit, based on comparison of AIC values.

Although not all versions of the MBSRQ-AS showed the same factorial structure as the original instrument, these adaptations presented satisfactory results regarding stability, reliability, and validity [46,47,48,50,52,54,56].

In the Chilean context, a study conducted with 451 participants between 15 and 25 years old evaluated, through exploratory factor analysis, the properties of the MBSRQ in a Chilean adolescent population, presenting sufficient internal consistency [57]. These findings confirmed that the MBSRQ factors have sufficient internal consistency to be used in research with the adolescent population in Chile. However, currently, there are no psychometric studies that have analyzed the factorial structure of the abbreviated version (MBSRQ-AS) in the Chilean population.

Based on the above, it is relevant to have instruments that allow dimensioning the different aspects linked to body image, not only in terms of satisfaction with the body but also with an emphasis on evaluation and orientation towards appearance, among other dimensions. The shortened version of the questionnaire (MBSRQ-AS) has generated much interest from researchers who focus on the appearance subscale of the MBSRQ and who prefer a shorter questionnaire. In addition, it is a widely used questionnaire in body image studies and stands out for the discrimination between orientation and appearance evaluation [52]. Additionally, it has been shown that the MBSRQ-AS is useful for studying body image in different age groups and sexes, with clinical and non-clinical samples [54].

Having this tool will allow having antecedents regarding body image and aspects of appearance, in some sectors of the normal Chilean adolescent population and its association with other psychological variables, such as self-esteem, self-concept, identity processes typical of the stage of development as well as psychosocial aspects of adolescence; and its relationship with psychopathological conditions, such as eating behavior disorders, thus identifying possible risk groups.

In line with these gaps and existing literature, the present study will address the following research questions:Does the proposed factorial structure (short version of the MBSRQ-AS) adjust to the characteristics of the Chilean adolescent and young non-clinical population?Does the dimensions of body image invariant hold by age, sex, and BMI?

## 2. Materials and Methods

### 2.1. Participants

The sample initially consisted of 645 cases, of which 31 (4.8%) did not have complete information on weight or height that would allow the calculation of BMI. After analyzing the distributions of the responses for the complete cases compared to those with missing values in BMI using the Kolmogorov–Smirnov test for independent groups, and finding no significative differences, it was decided to eliminate the 31 incomplete cases, since BMI was considered a central variable in the model to be evaluated.

The final sample was constituted of 614 Chilean adolescents and young people, of which 259 were male (42.2%) and 355 were female (57.8%). Participants were aged between 15 and 28 years, with an average of 18.81 years (SD = 2.46). They were residents of the Metropolitan Region, Santiago, and four regions from the coast and south-central zone of Chile. Participants were selected by non-probability sampling by quotas. The minimum size of the sample was determined according to Soper [58], considering α = 0.05, 1-β = 0.8, 5 latent variables, 34 observed variables, and an anticipated effect size of 0.16, yielding a minimum sample size for the model structure of 352 and a minimum size for detecting the effect size of 605 subjects. The formulas used to calculate the minimum simple size are available at https://www.danielsoper.com/statcalc/formulas.aspx?id=89 (accessed on 19 June 2022).

The Body Mass Index (kg/m^2^) showed an average of 22.5 (SD = 3.16).

The students’ participation in the study was voluntary and they did not receive any compensation for participating. The sociodemographic characteristics and health history reported by the participants and their family members are presented in Table 1.

### 2.2. Procedures

To recruit participants, schools and universities in five regions were contacted. Authorization was obtained from the institutions and participants signed an informed consent form to participate. In the case of minors, they had to sign an informed assent form and their parents signed an informed consent form. This procedure was supervised and approved by the Ethics Committee of Adolfo Ibáñez University.

Students answered the MBSRQ-AS and a sociodemographic questionnaire, shown in Table 1.

### 2.3. Multidimensional Body–Self Relations Questionnaire—Appearance Scales (MBSRQ-AS)

The MBSRQ-AS is a self-reported instrument that assesses only appearance-related aspects of body image. The MBSRQ-AS has 34 items divided into 4 subscales and items (1 to 22) are evaluated using a 5-point Likert scale and assess agreement from 1 (definitely disagree) to 5 (definitely agree). In the Body Areas Satisfaction (BAS) scale, different parts of the body are enumerated, assigning scores between 1 (very dissatisfied) and 5 (very satisfied). In the Self-Classified Weight (SCW) scale, participants are asked to classify their own weight from their own perspective and from the opinion of others; 2 items are rated between 1 (very underweight) and 5 (very overweight). Finally, item 23, which refers to the attempt to lose weight quickly through extreme diets, must be answered between 1 (never) and 5 (very often). To calculate the scores, the author [13] points out that the means of each scale must be calculated, after having inverted the score of 6 specific items (11, 14, 16, 18, 19 and 20).

In the original version, according to the author [41], in the female sample, Cronbach’s alpha ranged from 0.73 to 0.89 for the different subscales, and in the male sample, it ranged from 0.79 to 0.89. The one-month test–retest reliabilities of the MBSRQ-AS subscales ranged from r_tt_ =0.74 to r_tt_ = 0.91 in the female sample and r_tt_ = 0.79 to r_tt_ = 0.89.

In the Spanish version [54], Cronbach’s alpha ranged from 0.76 to 0.87. The convergent validity of this version was evaluated by Pearson correlation, with three factors of the Eating Attitude Test (EAT-26) [59], Eating Disorder Inventory-2 EDI-2) [60], Eating Disorder Examination Questionnaire (EDEQ) [61] also in the German version [46].

### 2.4. Statistical Analyses

The internal consistency for each subscale of the MBSRQ-AS was first analyzed using the reliability coefficient Cronbach’s Alpha, based on the Pearson correlation matrix and using the polychoric correlation matrix (Ordinal Alpha) as a reference to compare with previous studies. Since it is not possible to assume that the items are tau-equivalent, the McDonald’s Omega coefficient was calculated, also based on both matrices [62].

The analyzes were carried out with the software Mplus 8.5 [63]. The distributions of all the variables that would be included in the models were evaluated using the Kolmogorov–Smirnof (Lilliefords) test, rejecting the null hypothesis of univariate normality in all of them. Multivariate normality was tested with the Mardia test, obtaining skewness = 140.30, z = 14,357.57, *p* < 0.001 and kurtosis 1539.33, z = 40.58, *p* < 0.001; therefore, the null hypothesis was also rejected. Common method bias with Harman’s single factor was checked, obtaining a variance percentage of 20.737%, which indicates that the items do not present common method bias.

Considering the distributions of the variables and the quality of the ordinal data, it was decided to evaluate the fit of the theoretical model proposed by Cash using CFA with Weighted Least Square Mean and Variance Adjusted Estimators (WLSMV), which is robust for non-normal ordinal variables [64]. Model 1: one general factor; Model 2: five correlated factors.

The fit of the theoretical model was also evaluated by Exploratory Structural Equation Modeling (ESEM): a model with five correlated factors, using target rotation. This type of rotation allows the obtainment of the closest rotated solution to a pre-specified configuration, that is, the ESEM is used in a confirmatory mode [65], which in the present study corresponds to the five-factor model specified by the author. The decision to use ESEM is based on the fact that it provides greater flexibility than CFA since it does not force the possible cross-loads to be set to zero and generally yields more precise estimates of the correlations between the factors [65,66].

Due to the possible influence of age, sex, and BMI on the responses, a MIMIC (“multiple indicators, multiple causes”) model was evaluated. The MIMIC is used to test the invariance of the model, adding one or more covariates and examining their effects on the selected factors or indicators. These models, by involving only one input model and matrix at a time, have lower sample size requirements compared to multi-group invariance assessments in CFA. Another advantage is that the covariates used in the MIMIC models can also be continuous (not only categorical), as in the case of age and BMI in the present study [67].

Goodness-of-fit to the MIMIC model was obtained with sex-, age-, and BMI-correlated variables as covariates, formulated based on previous research and, in a second step, it was evaluated whether the items vary depending on these covariates [29,31,34,36].

The comparison of the fit of the models was based on χ^2^, Comparative Fit Index (CFI); Tucker–Lewis Index (TLI); Root Mean Square Error of Approximation (RMSEA) and Standardized Root Mean Square Residual (SRMR), using the criteria that indicate that values greater than 0.95 for CFI and TLI would account for an optimal fit and greater than 0.90 would be acceptable; for RMSEA, values under 0.06 would be considered optimal and under 0.08, acceptable. For SRMR, the criterion is values under 0.06 [68].

## 3. Results

As seen in Table 2, omega coefficients between 0.712 and 0.885 were obtained. They are not calculated for the SCW subscale because the model is not identified.

Regarding the evaluated models, it is observed (Table 3) that the model that shows the best fit is Model 3: five-factor ESEM: χ^2^ (401) = 1013.11, *p* < 0.01, CFI = 0.964, TLI = 0.950, RMSEA = 0.050, SRMR = 0.032.

Regarding the factor loads of the five-factor ESEM model, as shown in Table 4, it was found that all the items, except for item 5 (*I like my looks just the way they are*), which load 0.330 in AP and 0.399 in BAS, present the highest coefficient in the corresponding factor.

Regarding the magnitude of the coefficients, practically all the items show statistically significant factor loadings and >|0.3|, except item 11 (*I use very few grooming products*), which shows 0.115 in its AO factor (α = 0.05). This same item is the only one whose R^2^ is not statistically significant, which would be accounting for poor performance in relation to its scale (uniqueness > 0.9).

Regarding de composite reliability, coefficients of the subscales are > 0.7, except for OP (0.566), which shows lower factorial loads than the other subscales.

The average variance extracted (AVE) is less than 0.5 in all subscales except SCW (0.595), which is due to cross-loadings of an ESEM model.

MIMIC also obtains a good fit, χ^2^ (485) = 1187.75, *p* < 0.01, CFI = 0.961, TLI = 0.946, RMSEA = 0.049, SRMR = 0.033; showing significant paths between Sex with AO (0.323), BAS (−0.288), OP (0.262) and SCW (0.395); Age with AE (0.267), and BMI with AE (−0.388), BAS (−0.233), OP (0.263) and SCW (0.813). When evaluating the path of the items in relation to the covariates, three statistically significant variables were identified: item 12 (*I like the way I look without my clothes on*) (0.250) and item 32 (*How dissatisfied or satisfied are you with your weight*) regarding BMI (−0.227), and item 16(r) (*I don’t care what people think about my appearance*), regarding Age (0.163), as shown in Table 5 and Figure 1.

## 4. Discussion

The internal consistency obtained is consistent with the versions in Greek (α = 0.76 to 0.86) [48], Iranian (α > 0.75) [55], Brazilian (α = 0.73 to 0.90) [52], Spanish (α = 0.73 to 0.84) [51], French (α = 0.66 to 0.88) [47] and German (α = 0.78 to 0.90) [46]. Reliability has been estimated for each scale since the instrument is conceived as multidimensional, as the author points out: “Because this is a multidimensional assessment, researchers should not attempt to combine the various scales into a single measure” [41]. However, the CFA model of five correlated factors did not show an adequate goodness of fit. This lack of correct specification can be explained by the presence of multidimensional items that was evaluated using the ESEM model.

When evaluating the ESEM model, it is observed how the fit improves substantially. Thus, Model 3: five-factor ESEM shows better fit, which is consistent with the characteristics of the evaluated construct, while it is expected that the different components of body image are related to each other [1,69] but do not constitute a single dimension. The advantage, then, of using ESEM models is that they have allowed a better report of the structure in instruments that evaluate complex psychological constructs, which require considering multidimensionality [70,71].

From the MIMIC model, the results agree with previous research [9,19,72] that links sex, age, and BMI with components of Body Image.

Specifically for sex, women tend to be more concerned about their appearance and weight and have a more negative evaluation based on weight, which is consistent with numerous studies that indicate that young women overestimate their weight, compared to men [9,73,74], as well as less satisfaction with specific areas of their body [31,74]. This is consistent with results in the Brazilian adolescent female population [15], in addition to the relationship between the ideal of thinness and unsatisfactory body image [30,31].

Regarding BMI, worse appearance evaluation (AE), less satisfaction with specific areas of the body (BAS), greater concern about weight (OP), and worse weight-based evaluation are observed (SCW), as BMI increases, which is consistent with that indicated by Neumark–Sztainer [75] and with the findings of Durán-Agüero in Spain [76]. There is also a relationship between concern about weight (OP) and weight-based evaluation with BMI in several studies [77,78,79].

Age, on the other hand, was directly associated with a better appearance evaluation (AE), which could be attributed, according to the findings of Mellor et al. [29], to the fact that body dissatisfaction worsens in early adolescence.

Regarding the items that vary according to BMI, item 32: “*How dissatisfied or satisfied are you with your weight*”, is part of the BAS subscale. This subscale is inversely related to the BMI, that is, the higher the BMI, the lower the satisfaction with parts of the body, a relationship that the item also maintains with the covariate.

On the other hand, what is observed in item 12, which belongs to the AE subscale, is striking. This subscale shows an inverse relationship with BMI, that is, the evaluation of appearance worsens as the body frame increases. However, specifically in the evaluation of *how I look without my clothes on*, the relationship is inverse. Item 16 (r), *I don’t care what people think about my appearance*, which is part of the AO subscale, does not show differential function according to age; however, the item shows that this feature increases as age increases. These findings from the MIMIC model could be due to the large number of comparisons made, so this result may be an artifact or a capitalization of chance problem. It is necessary to deepen future studies on more specific aspects of body image (e.g., self-evaluation versus hetero-evaluation expectations as well as the influence of clothing on body dissatisfaction/satisfaction, among others), which allow us to account for the complexity of the construct.

As a limitation, it should be noted that MIMIC models do not allow the precision of multigroup CFA models to test invariance. However, they allow different variables to be evaluated jointly, which was adequate for the present study, since two of them are quantitative (age, BMI).

Other limitation of the study lie mainly in the fact that the BMI measurement was self-reported, which could reduce the accuracy of the data, since, as indicated by Allison et al. [80], it is expected that BMI and weight were underreported, and height was overreported in both genders, especially in obese individuals [81]. However, other studies indicate that self-reporting is not accurate for BMI prediction at the individual level, but it can be used as a simple and valid tool for estimating the BMI of overweight and obesity in epidemiological studies [82].

## 5. Conclusions

MBSRQ-AS replicates the five-dimensional structure in a non-clinical sample of young Chileans; however, indiscriminate inter-individual comparisons should be avoided since we have observed that the scores depend on sex, age, and BMI.

## Figures and Tables

**Figure 1 ijerph-20-00628-f001:**
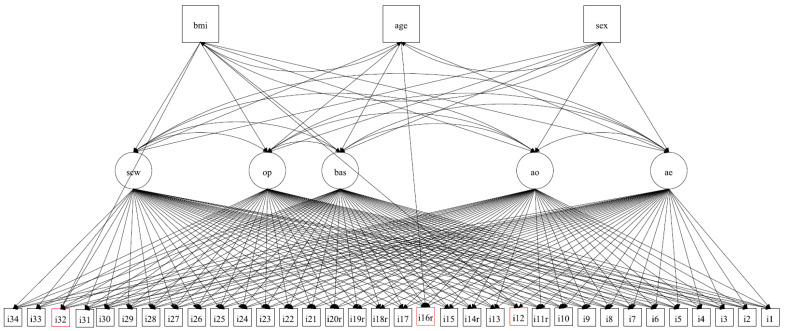
MIMIC model.

**Table 1 ijerph-20-00628-t001:** Sociodemographic characteristics.

Sociodemographic Characteristics	Frequency	Percentage
Total	614	100
Sex		
Male	259	42.18
Female	355	57.82
Age (years)		
15–17	189	30.78
18–20	259	42.18
21–23	149	24.27
24–26	15	2.44
27–28	2	0.33
Nutritional condition (WHO)		
Underweight	34	5.54
Normal	461	75.08
Overweight	105	17.10
Obesity	14	2.28
Education level		
Secondary	228	37.13
University	383	62.37
Missing values	3	0.49
Occupation		
Study	573	93.32
Study and work	36	5.86
Others	5	0.81
Health history		
Diabetes	12	1.96
Arterial hypertension	7	1.14
Family health history		
Overweight or obesity	325	53.02
Diabetes	276	45.10
Arterial hypertension	251	41.08

**Table 2 ijerph-20-00628-t002:** Descriptive Statistics and Internal Consistency for the MBSRQ-AS Scales (*n* = 614).

Scale	Items	M	SD	Skewness (SE)	Kurtosis (SE)	Mc Donald’s ω (95% CI)	Mc Donald’s ω Polychoric (95% CI)
Appearance Evaluation	7	3.322	0.799	−0.489	−0.051	0.87	0.885
				(0.099)	(0.197)	(0.855–0.886)	(0.871–0.899)
Appearance Orientation	12	3.609	0.612	−0.227	0.008	0.824	0.871
				(0.099)	(0.197)	(0.803–0.844)	(0.856–0.886)
Body Areas Satisfaction	9	3.412	0.679	−0.137	0.450	0.826	0.856
				(0.099)	(0.197)	(0.806–0.847)	(0.839–0.873)
Overweight Preoccupation	4	2.738	0.906	0.296	−0.483	0.712	0.781
				(0.099)	(0.197)	(0.674–0.749)	(0.753–0.809)
Self-Classified Weight	2	3.055	0.656	−0.150	−1.237	-	-
				(0.099)	(0.197)		

Note: M = mean; SD = standard deviation; SE = standard error; CI = confidence interval.

**Table 3 ijerph-20-00628-t003:** Models fit indices.

Model	χ^2^	df	CFI	TLI	RMSEA (90% CI)	SRMR	Meets Criteria
M1: one general factor	9175.23 **	527	0.498	0.465	0.163	0.158	No
					(0.161–0.166)		
M2: five factors	2365.03 **	517	0.893	0.884	0.076	0.078	No ^i^
					(0.073–0.079)		
M3: five factors ESEM	1013.11 **	401	0.964	0.950	0.050	0.032	Yes
					(0.046–0.054)		
MIMIC	1187.75 **	485	0.961	0.946	0.049	0.033	Yes
					(0.045–0.052)		

Note: χ^2^ = Chi-square; df = degrees of freedom; CFI = Comparative Fit Index; RMSEA = Root Mean Square Error of Approximation [90%CI]; SRMR = Standardized Root Mean Square Residual; ** *p* < 0.01. ^i^ Improper solution.

**Table 4 ijerph-20-00628-t004:** Factor loadings for 5 factors ESEM target rotation.

Factor	Appearance Evaluation			Appearance Orientation			Body Areas Satisfaction			Overweight Preoccupation			Self-Classified Weight			
Item	λ	S.E.	R^2^	λ	S.E.	R^2^	λ	S.E.	R^2^	λ	S.E.	R^2^	λ	S.E.	R^2^	Uniqueness
3	**0.887**	0.039	0.787	0.017	0.036	0.000	−0.121	0.042	0.015	0.073	0.041	0.005	0.055	0.032	0.003	0.365
5	**0.330**	0.043	0.109	−0.055	0.039	0.003	0.399	0.04	0.159	−0.229	0.042	0.052	−0.071	0.031	0.005	0.365
9	**0.767**	0.040	0.588	0.095	0.040	0.009	−0.089	0.045	0.008	0.062	0.049	0.004	0.042	0.035	0.002	0.475
12	**0.564**	0.040	0.318	−0.060	0.036	0.004	0.212	0.046	0.045	−0.046	0.038	0.002	−0.207	0.028	0.043	0.333
15	**0.307**	0.051	0.094	0.185	0.046	0.034	0.280	0.054	0.078	−0.052	0.053	0.003	−0.080	0.039	0.006	0.625
18 (r)	**0.544**	0.033	0.296	−0.044	0.034	0.002	0.285	0.035	0.081	−0.138	0.036	0.019	−0.148	0.025	0.022	0.246
19 (r)	**0.864**	0.041	0.746	−0.001	0.034	0.000	−0.045	0.044	0.002	0.074	0.037	0.005	0.007	0.031	0.000	0.324
1	0.020	0.051	0.000	**0.739**	0.035	0.546	−0.040	0.054	0.002	−0.108	0.047	0.012	0.021	0.038	0.000	0.504
2	0.198	0.050	0.039	**0.603**	0.035	0.364	−0.146	0.05	0.021	0.097	0.046	0.009	0.094	0.042	0.009	0.468
6	−0.162	0.049	0.026	**0.707**	0.037	0.500	0.063	0.051	0.004	−0.015	0.046	0.000	−0.068	0.039	0.005	0.550
7	−0.101	0.047	0.010	**0.833**	0.034	0.694	0.004	0.051	0.000	−0.203	0.042	0.041	0.005	0.037	0.000	0.447
10	−0.005	0.050	0.000	**0.669**	0.036	0.448	0.188	0.051	0.035	0.173	0.044	0.030	−0.072	0.036	0.005	0.423
11 (r)	0.035	0.073	0.001	*0.115*	0.056	0.013	−0.042	0.072	0.002	−0.055	0.063	0.003	0.084	0.048	0.007	*0.979*
13	0.014	0.053	0.000	**0.635**	0.039	0.403	0.018	0.052	0.000	−0.019	0.044	0.000	−0.104	0.039	0.011	0.613
14 (r)	0.068	0.055	0.005	**0.664**	0.041	0.441	−0.147	0.052	0.022	0.035	0.050	0.001	−0.043	0.037	0.002	0.508
16 (r)	−0.136	0.056	0.018	**0.417**	0.047	0.174	−0.122	0.057	0.015	0.145	0.053	0.021	−0.085	0.041	0.007	0.712
17	0.003	0.063	0.000	**0.527**	0.048	0.278	0.054	0.059	0.003	−0.073	0.053	0.005	0.019	0.042	0.000	0.746
20 (r)	0.103	0.052	0.011	**0.575**	0.039	0.331	0.029	0.055	0.001	0.170	0.043	0.029	−0.033	0.035	0.001	0.532
21	0.134	0.050	0.018	**0.418**	0.040	0.175	0.138	0.051	0.019	0.372	0.045	0.138	−0.003	0.039	0.000	0.518
26	0.249	0.050	0.062	0.064	0.041	0.004	**0.425**	0.059	0.181	−0.162	0.048	0.026	0.349	0.037	0.122	0.556
27	0.046	0.055	0.002	0.162	0.048	0.026	**0.421**	0.058	0.177	−0.123	0.057	0.015	0.329	0.041	0.108	0.722
28	0.082	0.057	0.007	−0.071	0.042	0.005	**0.574**	0.055	0.329	0.060	0.048	0.004	0.097	0.038	0.009	0.641
29	0.201	0.041	0.040	−0.081	0.034	0.007	**0.535**	0.041	0.286	−0.051	0.038	0.003	−0.270	0.027	0.073	0.300
30	0.074	0.052	0.005	−0.035	0.042	0.001	**0.622**	0.047	0.387	0.150	0.047	0.023	−0.009	0.036	0.000	0.584
31	−0.083	0.046	0.007	−0.077	0.035	0.006	**0.819**	0.044	0.671	0.229	0.044	0.052	−0.068	0.031	0.005	0.443
32	0.086	0.038	0.007	−0.025	0.032	0.001	**0.576**	0.042	0.332	−0.136	0.038	0.018	−0.316	0.024	0.100	0.279
33	−0.067	0.066	0.004	0.095	0.053	0.009	**0.380**	0.066	0.144	−0.136	0.059	0.018	0.156	0.042	0.024	0.859
34	0.213	0.036	0.045	0.077	0.030	0.006	**0.706**	0.038	0.498	−0.153	0.039	0.023	0.086	0.026	0.007	0.194
4	−0.036	0.049	0.001	0.230	0.046	0.053	0.057	0.049	0.003	**0.542**	0.049	0.294	0.008	0.037	0.000	0.549
8	−0.079	0.048	0.006	0.230	0.039	0.053	0.080	0.048	0.006	**0.559**	0.046	0.312	0.018	0.038	0.000	0.523
22	0.022	0.051	0.000	0.065	0.044	0.004	−0.122	0.051	0.015	**0.502**	0.057	0.252	0.272	0.038	0.074	0.499
23	−0.027	0.060	0.001	0.183	0.052	0.033	−0.240	0.055	0.058	**0.376**	0.063	0.141	0.183	0.045	0.033	0.524
24	−0.217	0.030	0.047	−0.035	0.031	0.001	0.015	0.032	0.000	0.168	0.034	0.028	**0.804**	0.033	0.646	0.141
25	−0.071	0.044	0.005	−0.071	0.040	0.005	0.050	0.042	0.003	0.141	0.040	0.020	**0.737**	0.034	0.543	0.397
CR	0.817			0.862			0.81			0.566			0.746			
AVE	0.420			0.364			0.334			0.250			0.595			

Note: (r) = reverse scored items; **Bold items** = Significant target loadings (*p* < 0.05); Underlined items indicate cross-loading items; S.E. = standard error; λ = Standardized factor loadings; CR = composite reliability (Omega); AVE = average variance extracted.

**Table 5 ijerph-20-00628-t005:** Standardized Coefficients.

	AE	AO	BAS	OP	SCW	Item 12	Item 32	Item 16(r)
Sex	−0.076	0.323 **	−0.288 **	0.262 **	0.395 **			
Age	0.267 **	−0.001	0.052	−0.006	−0.023			0.163 **
BMI	−0.388 **	0.017	−0.233 **	0.263 **	0.813 **	0.250 **	−0.227 **	

Note. AE = Appearance Evaluation; AO = Appearance Orientation; BAS = Body Areas Satisfaction; OP = Overweight Preoccupation, SCW = Self-Classified Weight; ** *p* < 0.01

## Data Availability

The data presented in this study are available on request from the corresponding author. The data are not publicly available due to privacy.

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
