# Peer review of "Psychometric Properties of the Multidimensional Body–Self Relations Questionnaire—Appearance Scales (MBSRQ-AS) in Chilean Youth"

_ijerph, 2022, doi:10.3390/ijerph20010628_

Round 1

Reviewer 1 Report

Reviewer’s Comments

11/09/2022

Issues in the Abstract Section

1-Please remove the texts from line 12 to the texts before “the aim of this paper” in line 17 on page 1 because these descriptions are details and should not appear in the abstract section. Also, the writing in this section did not hit the target. As the abstract of the present study, the authors should explicitly inform readers how many participants were used to validate the factor structure with general factor model, 5 factors model, and exploratory structural equation model of MBSRQ-AS scale, respectively. The results shows that blah, blah, blah…… Please rewrite abstract.

Issues in the Introduction Section

2-Similar to the above issue, the authors’ work did not catch major points. The authors could introduce the relationships among subscales because they should be correlated each other in fact. However, it should not be your focus here. In contrast, the authors should concentrate on how the adaptation, validation, and localization of this scale were accomplished in Chilean population. Although it is good that the authors thoroughly described how it was localized in different country, how this work was implemented and what the results were in Chile were not introduced and need to be intensified. Please rewrite this part as suggested.

3-The authors introduced the research purpose and focus from line 213 to line 218 on page 5. However, the reviewer does not see the research question(s) were formulated explicitly. Please raise your research questions in this section.

4-The reviewer noticed that the authors used ESEM in the present study, however, there is no information about its use in the adaption, localization, and validation of this scale with this quantitative method. Please do this work.

Issues in the Materials and Methods Section

5-The reviewer does not understand why there are three inconsistencies between “2. Materials and methods” and “3.1”, “3.2”, and “3.3” on page 5. Please explain this issue.

6-The author mentioned in line 221 on page 5 that there were 614 Chilean adolescents and young people in the present study. However, the information that the texts from line 234 to line 235 conveyed is completely inconsistent with the number “614 aforementioned. Please explain why this happened.

7-The same story occurred once again from line 231 to line 232 on page 5― the sum of the percentages here is not equal to 100%. Please explain what happened.

8-The authors used the wording “men” and “women” in line 22 on page 5. Do the authors think the writing is correct? Adolescents and young people ===Men and Women?  You’d better use “male” and “female” in the manuscript.

9- The authors used the wording “The age fluctuates” in line 22 on page 5. However, it is inappropriate. You could use ranged from instead or the students are aged between XX to XX.

10-In line 235, the authors provided the information of missing values. However, it cannot convince readers. The reason is very simple, there are several insistencies in the descriptive statistics of the sample. Also, the authors should introduce the rate and pattern of missing values. Is the pattern of missing values being MNAR or MCAR? Could you recompute the rate of missing values in the sample? Please do this work the reviewer just mentioned.

11-The authors did not provide enough information about the sample. In addition to the covariates such as age, sex, and BMI you mentioned in the manuscript, are there any other relevant demographic variables in your study? For example, is student economically disadvantaged? Please provide a table to present descriptive statistics of sample composition that include the information about sex, age, weight, height, BMI, occupation, educational level, health history of the parents, and relatives of the participants.

12-From line 226 to line 229, the authors introduced how they calculate minimum sample size. However, it is not enough, you still need to provide at least the formula used to compute sample size.

13-15-In line 230, the presentation of Body Mass Index is incorrect. The correct metric of this index should be “kg/m2.

14-The sentence from line 231 to line 232 has severe grammatical errors.

15-Below the subtitle “3.2 procedures”, the authors mentioned that with authorization of the institution to initiate the present research. However, it does not mean your research obtain the IRB approval letter and the consent letter from the participants. Please clarify this issue explicitly.

16-In the paragraph “3.2 Procedures”, the reviewer did not see the information about when or in what time period the research was done and whether the participants could quit the study at any stage if they want or they were rewarded with compensation for participating in the study. Please clarify blurred information.

17-The authors mentioned that it should not assume tau-equivalence condition holds for the items in line 247. However, a coefficient and ω coefficients were reported in Table 1, respectively. Now that you understand this very important rationale in psychometric theory, why did you do that? Please remove the characters “Cronbach’s Alpha” in line 246 and drop the columns a, a Low, and a High in Table 1 as well.   18-The reviewer noticed that it seems that there are no preliminary analyses. More specifically, did the authors investigate the sample for both univariate normality through the inspection of kurtosis and skewness values; and multivariate normality through Mardia’s two-side test of fit for skewness and kurtosis. For univariate normality, considering the guidelines of Finney & Distefano (2006) with the absolute value less than 2 of skewness and 7 of kurtosis as safe cutoff criteria in practice. To my knowledge, I am sure that Mplus software is capable of doing such tests. Please do this work.   19-How did the authors decide to use WLSMV in the present study? The reviewer is curious this issue. The authors need to explain why. Have you checked the outlier and extreme values?   20-The reviewer noticed that the authors did not provide any information about the MBSRQ-AS used in the present study. How many items were included in the study? What is the scoring rubric? For example, is the instrument that consists of 34 self-report items rated on a specific-point Likert scale ranging from 1 (e.g., “Every Day”) to specific point (“Seldomly”)? Also, what is the validity of the scales you used? (Convergent validity and discriminant validity) Furthermore, what is the item reliability? (Squared factor loadings) You need to provide information above.   

21-The most appropriate software, estimation method, rotation and procedure for the analysis should be determined reported. It is very weird that the authors mentioned Mplus version 8.50 was used to do analyses in line 260 for the present study because it should be reported in the beginning the paragraph. Once it is decided, an appropriate estimation method should be determined accordingly. By default, Mplus software employs Maximum Likelihood (ML) estimation method. If multivariate normality assumption is violated, then Maximum Likelihood with Robust (MLR) Standard Errors could be used for continuous indictors. However, additional steps should be taken for statistical model comparison as chi-squares cannot directly be compared for these estimation methods. (e.g., Satorra-Bentler scaled chi-square difference test which is also implemented in Mplus software with the DIFFTEST command for models with different estimators). For Bi-factor ESEM models with continuous indicators, the MLR estimator would be fine. As for the ESEM models with ordinal indicators, the WLSMV estimator should be used. Once the estimator has been chosen, then the most appropriate rotation method for ESEM should be designated and reported. Usually, there are three rotation methods to be considered depending on the purpose of your study: the 1st method is goemin rotation for exploratory approaches and reduce factor correlations maximally; the 2nd method is target rotations for confirmatory approaches; the 3rd method is orthogonal rotation for bifactor ESEM modeling. You can choose the most appropriate method for your case. Please do this work.

22-In Table 1 on page 6, the authors reported mean and standard deviation of the subscales. However, how did you get these numbers? These details should be described in texts. Also, I have to kindly remind you that the use of average score or sum score to convert the latent construct (subscale) to observed variable is inappropriate. It is obvious that neither you have the evidence nor the authority to claim that the weights between factor and observed indicators are equal. The correct way to convert latent factor to observed indicators is to compute factor score. You can google for help and there are a lot of resources you can reach out.

23-Please remove “Alpha coefficients between 0.706 and 0.865 and “in line 262 on page 6.

24-Please remove the row in which “M0: Null” is located. There is no need to present the null model at all in Table 2 on page 6.

25-In line 257, the authors used “and >0.90”. The reviewer cannot follow what does this mean.

26-In line 271, please remove “Esimator =” and the following characters and the word “Estimator” in line 272.

27-In line 271, line 280, and line 281, the authors used very strange symbols that the reviewer cannot follow—* = p < 0.05; ** = p <0 .01. It seems the document that the reviewer reviewed is a draft instead of a manuscript. Hope the authors can take your research work seriously. Thanks.

28-From line 279 to line 280, the authors made notes on the acronyms below Table 2. The format of the presentation there is not correct. The correct should look like as follows,

Note. AE = Appearance Evaluation; AO = Appearance Orientation; blah, blah…….

The similar issues appear in line 293 and line 294, please correct them.

29-The definition of Figure 1 on page 7 is very poor. Also, the layout is not suitable for beauty reason. Please adjust the scale of Figure 1 so that it is clearly visible to the reader. Meanwhile, please note its layout should be horizontal and it can fill the entire page.

30-Below line 292, Table 4 is presented about factor loadings for 5 factors ESEM model. You can choose to present it either horizontally or vertically. However, you have to place it on one page.

31-The reviewer does not understand what does the asterisk on the item mean in Table 4? Also, the reviewer does not know the meaning of 18(r), 19(r), 11(r), 14(R),16(R), and 20(R). Please explain them.

32-Now that M3: 5 factors ESEM is the optimal model, why did the authors run a MIMIC ESEM model that include covariates age, sex, and BMI? Although the data-model fit indices are better than 5 factors ESEM model, it does not make sense to directly test measurement invariance on MIMIC model. You never test if measurement invariance on the original measurement model holds, which looks like a scale never calibrated. So, please do the measurement invariance test on 5 factors ESEM model.

33-Please rewrite Discussion and “Conclusion” sections after correcting all problems above.

Reviewer 2 Report

The present manuscript reports on a psychometric study of the properties of an adapted version of the MBSRQ questionnaire in Chile. The study is an interesting one, the sample size seems sufficient to answer the research questions, and the analyses are very advanced. Nevertheless, I would like to offer feedback in order to improve the quality of the paper further.

1. Citations should be removed from the abstract completely.

2. I suggest to include a paragraph or two outlining the negative implications of body image on mental health. This would strengthen the argument of why should one care about the validation of this instrument.

3. It would also be interesting to see the importance of administering a short version of the measure, too.

4. In lines 185-187, the authors mention some indices but I am not sure which indices they refer to.

5. How was this scale scored? what type of measurement scale was used?

6. Although I agree with the selection of the WLSMV estimator in principle, I wish to also see some references backing up this choice.

7. If the items were scored with an ordinal scale, then have the alpha and omega coefficients been estimated based on polychoric correlations? Which omega coefficient was selected (total or hierarchical)?

8. It would be better to specify that omega lower vs. upper are the upper and lower bounds of the confidence interval.

9. Why select an exploratory SEM model instead of the classical EFA model?

10. Can the authors also provide a correlation matrix in table 1, too?

11. I think the ESEM model is not that theoretically and practically meaningful. For example, we cannot actually score the scale if all items have freely estimated cross-loadings. This has significant implications for practice since the latent factors do not make sense now given the cross-loadings. For instance, item 15 has statistically significant cross-loadings on nearly every latent factor. Therefore, I am not sure that we can actually define these latent factors based on the theoretical underpinnings of this instrument. The authors could write more to convince the substantive reader.

12. Has the MIMIC model been specified correctly? From the figure 1, I am not sure about this. The items should also be regressed on the exogenous covariates (i.e., age, gender, bmi) (see Brown, 2015).

13. It would be interesting to see whether a bi-factor or second-order factor model is better.

Round 2

Reviewer 1 Report

Reviewer’s Comments

12/20/2022

1-The reviewer described the research questions in the following manner, which would be very straightforward and easily to follow,

In line with these gaps and existing literature, the present study will address the following research questions:

1-Does the proposed factorial structure (short version of the MBSRQ-AS) adjust to the characteristics of the Chilean adolescent and young non-clinical population?

2-Does the dimensions of body image invariant hold by age, sex, and BMI?

If the authors can stand in reader’s position more, think feeling for them, and put yourself into their shoes, in other words, if you were a reader, would you be willing to read a research paper without a clear research question?

2-Please remove the texts from line 233 to line 238 due to the redundancy.

3-The 2nd and the 3rd column in Table 1 shall be changed “Frequency” and “Percentage”, respectively. Also, the decimal places shall be aligned in Table 1 as well.

4-The reviewer noticed that there are many places both in the revised manuscript and the authors response saying why the alpha coefficients were used and computed. Because the reviewer has pointed out this issue (tau-equivalence assumption) several times and the author also knew this assumption is not easily to be satisfied in real practice. Please remove all the description of Cronbach’s alpha coefficients and the values in the manuscript. So, just calculate and keep omega coefficients instead.

5-Please test tau-equivalence assumption for sure. Otherwise, it is unreasonable to present the results of Cronbach’s alpha coefficient. When comparing the congeneric and tau-equivalent models, the question is whether constraining the factor loadings in the tau-equivalent model significantly degrades the model fit. If it does, then the assumption of tau-equivalence does not hold. We can use an LR or -difference test to compare the overall fit of the two models. These tests are valid because the congeneric and tau-equivalent models are nested models. Specifically, the tau-equivalent model is nested within the congeneric model. The authors can compute the difference of the chi-squared value and degree of freedom between the congeneric model and the tau-equivalence model, and p-value. If the resultant p-value is less than 0.05, it means that the null hypothesis got rejected.

6. From line 339 to line 340, the authors mentioned that reliability coefficients were not calculated because there were 2 items there. Please refer to this resource at  https://www.youtube.com/watch?v=RZMrlt1pKNs. Although this process is done with AMOS software, the procedure and the rationale still apply to Mplus software. Please do this work.

7-As the baseline model (5 factor CFA model), how did the authors check the discriminant validity and convergent validity? If the authors do not know how to compute them, please refer to the Table 1 in this resource at https://iopscience.iop.org/article/10.1088/1742-6596/890/1/012163/pdf

8-As the baseline model (5 factor CFA model), how did the authors evaluate the construct validity?

9-After model-fit is established, measurement quality needs to be confirmed. The authors should report item reliability (the squared factor loadings), scale reliability (composite reliability). In addition, please inspect the standardized factor loadings (lambda >.35), the item uniqueness (e.g., residual error variances >.10 but <.90).

10-In line 345, the authors mentioned that the best model fit is Model 3…. Where are the other evaluated models? What is model 1, model 2, and model 4, and model 5? The authors still need to add a table to present the results of multiple data-model fits of those models.

11-The authors should check common method bias. However, this very important step is missing. If the authors do not know how to do this, please refer to this resource at https://www.youtube.com/watch?v=0GRob-VMPFM or with Harman's single factor test at https://www.youtube.com/watch?v=-RhxlC_3VIE.

12- In the 1st version of the manuscript, the reviewer noticed that there was a model called “general factor model”. Does this mean M1 is bi-factor model? If it is the case, the authors can combine this model with the reviewer’ comment 9 together. Meanwhile, the reviewer is curious why there is no second-order CFA model as a competing model when determining the optimal CFA model?

13-After obtaining the optimal model (5 factor ESEM model) among a series of competing models, the authors need to check whether there are some competing ESEM models. In other words, how did the authors justify the 5-factor ESEM model is the finalized model? According to the relevant theory in this field, there might be several competing ESEM models as follows to compare with the 5-factor ESEM model,

1) a 5-factor second-order ESEM model;

2) a bi-factor ESEM model;

3) a correlated 5-factor ESEM-within-CFA model.

If the authors do not know how to run these models with Mplus, you can refer to this resource at https://www.statmodel.com/download/Webnote%20-%20Hierarchical%20Exploratory%20Structural%20Equation%20Model.pdf or other resources available to you via google.

14-The authors used MIMIC-ESEM model to test measurement invariance by several demographic variables. The reviewer does not think that this method is wrong, however, it has unconquerable limitations—this method can just address intercept invariance and latent factor mean invariance. The reviewer suggests the authors refer to this resource at https://www.youtube.com/watch?v=Z4A-_UpVcTY for further reference.

15-If the authors decided to keep using MIMIC ESEM model to test invariance, please ignore previous reviewer’s comment. One thing the authors failed to test is the interaction term between the demographical variables. For example, age×gender and/or bmi×gender. Frankly, the implementation of this method is much more complicated than the method that appear in the previous reviewer’s comment. Either of these methods can be chosen at the authors’ discretion.

16-Please rewrite the section of discussion after finishing running the above analysis.

17-Please update the statistical analysis part in the manuscript according to the revisions the authors made this time.
